# Incidence of deep vein thrombosis through an ultrasound surveillance protocol in patients with COVID-19 pneumonia in non-ICU setting: A multicenter prospective study

Filippo Pieralli[1]☯*, Fulvio Pomero[2]☯, Margherita Giampieri[1], Rossella Marcucci[3], Domenico Prisco[3], Fabio Luise[1], Antonio Mancini[1], Alessandro Milia[1], Lucia Sammicheli[1], Irene Tassinari[1], Francesca Caldi[1], Francesca Innocenti[1], Antonio Faraone[4], Chiara Beltrame[4], Riccardo Pini[1,3], Andrea Ungar[1,3], Alberto Fortini[4]

**1** COVID-19 Intermediate Care Unit, Careggi University Hospital, Florence, Italy, **2** Internal Medicine COVID-19 Unit, Ospedale Michele and Pietro Ferrero, Verduno (Cuneo), Italy, **3** Department of Clinical and Experimental Medicine, University of Florence and Careggi Hospital, Florence, Italy, **4** Internal Medicine COVID-19 Unit, San Giovanni di Dio Hospital, Florence, Italy

☯ These authors contributed equally to this work.
* filippopieralli@gmail.com

**Data Availability Statement:** All relevant data are within the paper.

## Abstract

### Objective

The aim of this study was to assess the incidence of deep vein thrombosis (DVT) of the lower limbs, using serial compression ultrasound (CUS) surveillance, in acutely ill patients with COVID-19 pneumonia admitted to a non-ICU setting.

### Methods

Multicenter, prospective study of patients with COVID-19 pneumonia admitted to Internal Medicine units. All patients were screened for DVT of the lower limbs with serial CUS. Anticoagulation was defined as: low dose (enoxaparin 20–40 mg/day or fondaparinux 1.5–2.5 mg/day); intermediate dose (enoxaparin 60–80 mg/day); high dose (enoxaparin 120–160 mg or fondaparinux 5–10 mg/day or oral anticoagulation). The primary end-point of the study was the diagnosis of DVT by CUS.

### Results

Over a two-month period, 227 consecutive patients with moderate-severe COVID-19 pneumonia were enrolled. The incidence of DVT was 13.7% (6.2% proximal, 7.5% distal), mostly asymptomatic. All patients received anticoagulation (enoxaparin 95.6%) at the following doses: low 57.3%, intermediate 22.9%, high 19.8%. Patients with and without DVT had similar characteristics, and no difference in anticoagulant regimen was observed. DVT patients were older (mean 77±9.6 vs 71±13.1 years; p = 0.042) and had higher peak D-dimer levels (5403 vs 1723 ng/mL; p = 0.004). At ROC analysis peak D-dimer level >2000 ng/mL (AUC

**Funding:** The authors received no specific funding for this work.

**Competing interests:** The authors have declared that no competing interests exist. On behalf of all coauthors the corresponding author. Filippo Pieralli MD

0.703; 95% CI 0.572–0.834; p = 0.004) was the most accurate cut-off value able to predict DVT (RR 3.74; 95%CI 1.27–10, p = 0.016).

## Conclusions

The incidence of DVT in acutely ill patients with COVID-19 pneumonia is relevant. A surveillance protocol by serial CUS of the lower limbs is useful to timely identify DVT that would go otherwise largely undetected.

## 1. Introduction

After the initial COVID-19 outbreak in China, Italy was the most affected European country in the earlier phase of the pandemic, and faced an unprecedented health-care emergency [1]. The SARS-CoV-2 infection pandemic is generating a heavy burden of morbidity, mortality, and stress for the health-care systems. An emerging issue, especially in moderate to severe cases of COVID-19, is the significant prevalence of a "SARS-CoV-2 coagulopathy" [2, 3] characterized by elevated D-dimers, and other biohumoral markers of activation of coagulation.

Along with the documentation of coagulopathy, an increased incidence of venous thromboembolism (VTE) has been reported in patients hospitalized for COVID-19 pneumonia with respect to "usual" infectious respiratory disease. Actually, the incidence of VTE in non-ICU patients admitted for infectious respiratory disease is estimated around 5%, with adequate thromboprophylaxis, while the incidence in COVID-19 patients has been reported as high as 20% [4–11]. In the SARS-CoV-2 pandemic scenario, the most reliable data derive from studies carried out in ICU patients, while less robust information is available for patients with COVID-19 pneumonia admitted to non-ICU settings, such as those of Pneumology and Internal Medicine Units (IMUs).

To investigate this unresolved issue, we designed a prospective study, using serial ultrasonographic surveillance in order to assess the incidence of deep vein thrombosis (DVT) of the lower limbs in a cohort of patients with COVID-19 admitted to IMUs.

## 2. Patients, setting, and methods

This is a prospective cohort study carried out at three large Italian hospitals. The COVID-19 Intermediate Care Unit of the Careggi University Hospital, the Internal Medicine COVID-19 Unit of the San Giovanni di Dio Hospital, both in Florence, and the Internal Medicine COVID-19 Unit, Ospedale Michele and Pietro Ferrero, Verduno (CN). Between March 21st and May 25th, 2020, all consecutive patients aged more than 18 years with a definitive diagnosis of SARS-CoV-2 pneumonia admitted to these units were enrolled. The diagnosis of SARS-CoV-2 infection was made by real-time reverse transcription polymerase chain reaction (RT-PCR) assay on naso-pharyngeal swab and/or bronchoalveolar lavage. On admission, thromboprophylaxis with enoxaparin or fondaparinux was prescribed to all patients according to protocols in use in the hospitals, except to those with absolute contraindications and to those with indication to full anticoagulation. According to literature criteria [12] the doses of initial anticoagulation were defined as follows:

- Low dose anticoagulation equivalent to enoxaparin 20–40 mg or fondaparinux 1.5–2.5 mg per day.

- Intermediate dose anticoagulation equivalent to enoxaparin 60–80 mg per day.

- High dose anticoagulation equivalent to enoxaparin 120–160 mg or fondaparinux 5 to 10 mg per day, or oral anticoagulation with vitamin K antagonists (VKA) or direct oral anticoagulants (DOACs).

All patients were screened for DVT of the lower limbs with serial color-coded Doppler and compression ultrasonography (CUS) within 72 hours since admission and subsequently at 5–7 day intervals, and before discharge. Ultrasonography examination was performed by experienced physicians with long-standing expertise in vascular ultrasonography. The lower limb veins were explored with a linear 7.5 MHZ probe in B-mode and using Color-Doppler mapping and compression technique with transverse and longitudinal scan with the following machines: Canon Viamo c100, Esaote MyLab 60, Esaote MyLab 70 and Philips CX50. The ultrasound scan was performed along the proximal femoral and popliteal district bilaterally, by a three-point examination of the common and superficial femoral veins, and the popliteal veins [13], and if possible was extended to the distal infrapopliteal vein district. The lack of vein compression was the only diagnostic criterion for the diagnosis of DVT. After receiving diagnosis of DVT, patients were eventually shifted to full dose anticoagulation. DVTs were considered asymptomatic or symptomatic in relation to the absence or presence of clinical signs (such as, leg swelling, pain, or both) suggesting a venous thrombosis. A multidetector computed tomographic pulmonary angiography (CTPA) was ordered in those cases with clinical suspicion of pulmonary embolism (PE) whenever indicated by the physician in charge, and no predefined protocol was in place for the indication to the execution of CTPA.

Demographic, clinical, and laboratory variables were recorded. Specifically, blood samples to assess blood cell count, aPTT, PT/INR, fibrinogen, C-reactive protein (CRP), and D-dimers (expressed as ng/mL FEU–fibrinogen equivalent unit) were obtained on admission and at 48 to 72 hours intervals. The method and the laboratory normal values of D-dimers were the same for all the three hospitals' laboratories with an upper reference limit of 500 ng/mL FEU. For the purposes of this study "peak D-dimers" was defined as the highest value during hospital stay for patients without DVT and the highest value at the time (±24h) DVT was diagnosed for the DVT group. The Padua Prediction Score (PPS) [14], a scoring system which has been validated for the prediction of VTE risk in hospitalized patients was calculated for every patient at admission (>4 points defines high risk). All major hemorrhagic complications were recorded; they were defined as major bleeding leading to death, hemodynamic instability, need for blood transfusions, or occurring at any critical site.

The primary end-point of the study was the incident diagnosis (cumulative incidence) of DVT of the lower limbs by CUS. The secondary end-point was the identification of risk factors for DVT.

Human beings were involved in the research, the study protocol was approved by the ethics committee of the coordinating center, Azienda Ospedaliera Universitaria Careggi, Florence, Italy (COCORA protocol 17104), and was performed in agreement with the principles set in the Declaration of Helsinki. Informed consent was not needed, since point-of-care ultrasonography is a standard of care for the evaluation and monitoring of patients with COVID-19; only signed consent for personal data collection and treatment was requested; all the data were analyzed anonymously.

## 3. Statistical analysis

Continuous variables were expressed as mean ± standard deviation or as median and interquartile ranges (IQR; 25th–75th percentile) as appropriate. In general, statistical comparisons were performed using Student's t test and one-way ANOVA models for the comparison of continuous normally distributed variables and Mann-Whitney $U$ test for continuous not

normally distributed variables. The Chi-square test or Fisher's exact test were used for the comparison of categorical variables. A receiver-operating-characteristic (ROC) curve analysis was used to obtain the most accurate D-dimers peak cut-off for the prediction of DVT. Risk ratio (RR) of variables and 95% confidence intervals (CIs) were calculated using univariate and multivariable logistic regression analysis. All p values were two-tailed and considered significant when <0.05 (95% CI). All analyses were performed using SPSS statistical software 21.0 (SPSS, Chicago, Illinois, USA).

## 4. Results

Two-hundred-twenty-seven patients constituted the entire cohort of the study (63 patients at Careggi Hospital, 79 patients at Michele e Pietro Ferrero Hospital, and 85 patients at S. Giovanni di Dio Hospital, respectively). Clinical, demographic characteristics and laboratory values of the general population are reported in Table 1. There was a slight prevalence of males (57%), and elderly patients (57% with age > 70 years). There was a high prevalence of hypertension and other cardiovascular diseases; diabetes was present in nearly 1 out of 4 patients. All patients were considered at high risk (>4) at PPS, and all patients had a score ≥5; specifically 87% of patients had a PPS of 5 and 6 points. All patients had moderate-severe pneumonia with respiratory failure requiring oxygen therapy, and 36 patients (15.8%) received high flow oxygenation with nasal cannulae and/or non invasive mechanical ventilation by helmet or mask.

Overall, during the hospitalization DVT was diagnosed by CUS in 31 patients corresponding to an overall incidence of 13.7%. In 14 patients (6.2%) DVT was proximal, while in the remaining 17 patients (7.5%) it involved veins of the infrapopliteal district; overall 4 patients had bilateral DVT. All patients received CUS of proximal veins, while in 117 patients (51.5%) the ultrasonographic exploration was extended to the infrapopliteal district. Only 2 patients had symptomatic DVT, 1 popliteal and 1 femoral, all the remaining cases (29/31 patients, 94%) were asymptomatic. The timing of DVT detection from admission was within 72 hours in 18 patients (58.1%), between day 5 to 8 in 11 patients (35.5%), and between day 12 to 15 in 2 patients (6.4%). Overall, 9 patients (3.9%) were diagnosed with PE at CTPA. Of these, 5 had a concomitant DVT of the lower limbs, while in 4 cases PE was isolated. CTPA was available in 67 out of 227 patients (29.5%).

All patients received anticoagulation: enoxaparin was the most used anticoagulant drug (217 patients, 95.6%), while only a minority received fondaparinux (5 patients), or VKA/DOACs (5 patients) for concomitant clinical indications to receive full-dose anticoagulation.

The categories of initial anticoagulant treatment were as follows: low dose (thromboprophylaxis) 130 patients (57.3%), intermediate dose 52 patients (22.9%), and high dose 45 patients (19.8%). There was no difference in DVT occurrence in the three different categories of anticoagulation (Table 2). Patients who experienced DVT received lower mean daily dose of enoxaparin compared to patients without DVT ($47\pm23$ mg/day vs $66\pm38$ mg/day; p = 0.042, Table 2). No influence of body weight over the choice of enoxaparin doses was evident: the mean daily dose of enoxaparin was $59\pm30$ mg in patients with BMI>30 Kg/m$^2$ vs $63\pm37$ mg in patients with BMI ≤30 Kg/m$^2$, respectively (p = 0.672).

Patients with and without DVT had similar clinical characteristics, laboratory values and Padua Prediction Score; they differed only for age and peak levels of D-dimers (Table 2). Notably, patients diagnosed with DVT were older (age >70 years 83.3% vs 52%; p = 0.028) and with significantly higher levels of the median peak D-dimers (5403 vs 1723 ng/mL; p = 0.004). When the D-dimer level on hospital admission was considered, no difference was observed in

**Table 1. Demographic, clinical characteristics and laboratory findings and outcomes of the entire cohort of patients.**

| | Total number of patients = 227 |
|---|---|
| Age years (mean±SD) | 72±13 |
| Age> 70 years, n (%) | 127 (57) |
| Male gender, n (%) | 129 (56.8) |
| **Clinical features** | |
| Hypertension, n (%) | 129 (56.8) |
| Cardiovascular disease, n (%) | 77 (33.9) |
| Diabetes, n (%) | 55 (24.2) |
| COPD, n (%) | 37 (16.2) |
| Obesity (BMI>30 Kg/m$^2$), n (%) | 25 (11) |
| Cancer, n (%) | 20 (8.8) |
| **Main laboratory findings** | |
| Platelet count, $10^9$/L (mean±SD) | 220567±96698 |
| White Blood Cell, $10^9$/L (mean±SD) | 8.78±11.87 |
| Creatinine, mg/dL (mean±SD) | 1.33±1.29 |
| PT, activity % (mean±SD) | 72.3±15.5 |
| aPTT, seconds (mean±SD) | 36.6±45.1 |
| Fibrinogen, mg/dL (mean±SD) | 599±166 |
| C reactive protein, mg/L, (median—IQR) | 16.5 (6.7–56.5) |
| D-dimer value on admission, ng/mL (median–IQR) | 1483 (576–7732) |
| D-dimer peak value, ng/mL (median—IQR) | 1865 (1046–5682) |
| D-dimer peak value 1st quartile, ng/mL | < 1046 |
| D-dimer peak value 2nd quartile, ng/mL | 1046–1865 |
| D-dimer peak value 3rd quartile, ng/mL | 1866–5682 |
| D-dimer peak value 4th quartile, ng/mL | >5682 |
| **Padua Prediction Score** | |
| 5, n (%) | 79 (34.8) |
| 6, n (%) | 119 (52.4) |
| >6, n (%) | 29 (12.8) |
| Mean±SD | 5.91±1.0 |
| **Diagnosis of DVT** | |
| Total, n (%) | 31 (13.7) |
| Proximal, n (%) | 14 (6.2) |
| Distal, n (%) | 17 (7.5) |
| **Timing of DVT diagnosis since admission** | |
| Day 0–3, n (%) | 18 (58.1) |
| Day 5–8, n (%) | 11 (35.5) |
| Day 12–15, n (%) | 2 (6.4) |
| **Outcome measures** | |
| In-hospital length of stay, days (mean±SD) | 12.2±6.7 |
| Step up to ICU, n (%) | 12 (5.3) |
| In-hospital death, n (%) | 22 (9.7) |
| Step up to ICU or in-hospital death, n (%) | 34 (15) |

COPD: Chronic obstructive pulmonary disease; PT: prothrombin time; aPTT: activated partial thromboplastin time; DVT: deep vein thrombosis; ICU: Intensive Care Unit.

**Table 2. Demographic, clinical characteristics and laboratory findings and outcomes of patients with e without DVT.**

| | DVT (n = 31) | No DVT (196) | *p* |
|---|---|---|---|
| Age, years (mean±SD) | 77±10 | 71±13 | 0.042 |
| Age> 70 years, n (%) | 25 (83) | 102 (52) | 0.028 |
| Male gender, n (%) | 20 (66) | 109 (56) | 0.341 |
| **Clinical features** | | | |
| Hypertension, n (%) | 16 (51.6) | 113 (57.6) | 0.629 |
| Cardiovascular disease, n (%) | 10 (32.2) | 67 (34.2) | 0.453 |
| Diabetes, n (%) | 9 (29.0) | 46 (23.5) | 0.771 |
| COPD, n (%) | 5 (16.1) | 32 (16.3) | 0.532 |
| BMI>30 Kg/m$^2$, n (%) | 4 (12.9) | 21 (10.7) | 0.694 |
| Cancer, n (%) | 3 (9.6) | 17 (8.7) | 1.0 |
| **Main laboratory findings** | | | |
| Platelet count, 109/L (mean±SD) | 237800±96380 | 217875±96846 | 0.393 |
| White Blood Cell, $10^9$/L (mean±SD) | 9.62 ± 4.90 | 8.65 ± 12.62 | 0.736 |
| Creatinine, mg/dL (mean±SD) | 1.7±1.4 | 1.3±1.2 | 0.123 |
| PT, activity % (mean±SD) | 67.5±19.3 | 73.0±14.7 | 0.137 |
| aPTT, seconds (mean±SD) | 32.5±12.1 | 30.9±5.4 | 0.527 |
| Fibrinogen, mg/dL (mean±SD) | 645.1±167.1 | 591.9 ± 165.6 | 0.185 |
| C reactive protein, mg/L, (median-IQR) | 12.0 (5.7–27.6) | 16.5 (6.9–75.2) | 0.322 |
| D-dimer peak value, ng/mL (median-IQR) | 5403 (1751–34260) | 1723 (1008–4729) | 0.004 |
| D-dimer value on admission,ng/mL (median-IQR) | 2349 (665–11100) | 1832 (687–13174) | 0.683 |
| **Padua Prediction Score** | | | |
| 5–6 points, n (%) | 26 (83.9) | 172 (87.7) | 0.931 |
| Points (mean±SD) | 6±0.86 | 5.9±1.03 | 0.677 |
| **Outcome measures** | | | |
| In-hospital stay, days (mean±SD) | 12.1±8.5 | 11.8±8.7 | 1.0 |
| Step up to ICU, n (%) | 2 (6.4) | 10 (5.1) | 0.861 |
| In-hospital death, n (%) | 4 (12.9) | 18 (9.2) | 0.478 |
| Step up to ICU or in-hospital death, n (%) | 6 (19.3) | 28 (14.3) | 0.431 |
| **Intensity of anticoagulation** | | | |
| Low dose, n (%) | 18 (58.1) | 112 (57.2) | 0.948 |
| Intermediate dose, n (%) | 7 (22.6) | 45 (22.9) | 1.0 |
| High dose, n (%) | 6 (19.3) | 39 (19.9) | 0.954 |
| Enoxaparin daily dose mg, (mean±SD) | 47±23 | 66±38 | 0.004 |

DVT: deep vein thrombosis; COPD: Chronic obstructive pulmonary disease; PT: prothrombin time; aPTT: activated partial thromboplastin time; ICU: Intensive Care Unit.

patients who subsequently were diagnosed with (median 2349 ng/mL [665–11100]) and without DVT (median 1832 ng/mL [687–13174]) (p = 0.683, Table 2).

At ROC analysis the most accurate cut-off of peak D-dimers to identify patients with DVT was 2000 ng/mL (AUC 0.703; 95% CI 0.572–0.834; p = 0.004, sensitivity 75%, specificity 65%, negative predictive value 93%, positive predictive value 21%, accuracy 58%). At univariate analysis, patients with age > 70 years and D-dimers peak levels greater than 2000 ng/ml had an increased risk of DVT, (RR 3.64, 95% CI 1.15–11.5; p = 0.028, and RR 3.74, 95% CI 1.28–10.9; p = 0.016, respectively). At multivariable analysis, only D-dimers peak levels greater than 2000 ng/ml retained their predictive value for DVT diagnosis (RR 3.74; 95%CI 1.27–10, p = 0.016). Two patients experienced major hemorrhagic complications. One patient without

DVT treated with prophylactic high-dose anticoagulation (enoxaparin 160 mg per day) had a large retroperitoneal muscular hematoma requiring blood transfusions and drainage. Another patient, without DVT while on treatment with high-dose enoxaparin (120 mg per day) had a bleeding from gastric ulcer, requiring endoscopy treatment and blood transfusions. No difference was found in mean duration of hospital stay, need of step up to ICU, in-hospital mortality, and the combination of the last two in patients with and without DVT (Table 2).

## 5. Discussion

In this multicenter study, a surveillance protocol with systematic ultrasonography of the lower limbs in patients admitted to IMUs with COVID-19 pneumonia showed an overall incidence of DVT of 13.7%, 6.2% in proximal (6.2%) and 7.5% in distal venous district. To the best of our knowledge our study, including 227 patients screened for DVT, is the largest prospective, multicenter study in a non-ICU setting. Of note 94% of DVTs were asymptomatic, which means that they had been undetected without a serial ultrasound protocol of surveillance [15]. All patients received anticoagulation, at least at doses indicated for VTE prophylaxis in medical patients at high risk as indicated by a Padua prediction score >4. Specifically, 57% of patients received low dose anticoagulation, while the remaining 43% received intermediate or high dose anticoagulation. These data taken together suggest a significant incidence of lower limb DVT in patients with COVID-19 pneumonia in non-ICU setting despite anticoagulation.

This prevalence is in accordance with the results of the recent meta-analysis by Jimenez et al [16] that reported an overall estimated pooled incidence of VTE of 17%. Demelo-Rodriguez and coworkers in a single-center study on patients hospitalized in non-intensive care units with COVID-19 pneumonia and D-dimers > 1000 ng/ml, described a incidence of DVT of 14.7%, of whom only 1 (0.6%) were proximal [17]. Avruscio et al, in a series of 44 patients admitted to medical wards in Padua, Italy, described an overall occurrence of DVT of 22.7%; when the observation was limited to the lower limbs the prevalence of DVT was 13.6%, of which 9.1% were distal and 4.5% proximal [18]. A recent study from a single center in Rome (Italy) on 84 patients with COVID-19 reported an incidence of DVT detected by systematic ultrasonography screening of 11.9% with higher incidence of distal (9.5%) versus proximal (2.4%) DVT [19].

In our study, a feature that differed substantially from the previous ones is the higher incidence of proximal DVT compared to distal. A possible explanation could be that in this study the extension to the distal venous district was not mandatory per protocol, and only 51.5% (117/227 patients) underwent US extended to the infrapopliteal district. Then, we can assume that if CUS had been extended to the distal veins in all patients, the diagnosis of distal DVT would have been even higher, indeed increasing the incidence of distal and overall DVT. In general, the prevalence and incidence of DVT in COVID-19 patients features much higher when compared to that of DVT in medically ill non COVID-19 patients admitted to medical wards. In PREVENT trial, which compared dalteparin 5000 IU daily to placebo for the prevention of venous thromboembolism in acutely ill medical patients, the incidence of proximal DVT mainly assessed by ultrasonography was 2% in the active treatment arm and 4.05% in the placebo group [20]. Indeed, the incidence of DVT in COVID-19 patients on anticoagulant therapy is much higher than the incidence of hospitalized medically ill non COVID-19 patients not receiving prophylactic treatment. The very high prevalence of thrombotic events in COVID-19 patients, despite adequate thromboprophylaxis, could be explained by the overwhelming inflammatory host response with cytokines induced endothelial damage and to the direct involvement of endothelial cells by SARS-CoV-2 [21, 22].

A finding that is worth to note is the timing of detection of DVT by CUS surveillance in our study. In 58% of patients DVT was diagnosed within 72 hours from admission, in 35.5% between day 5 and 8, and in a very limited number of cased in the late course of hospital stay. Very few data exist on timing of DVT diagnosis in COVID-19 patients in current literature. In a retrospective study from three New York City hospitals on a cohort of ICU patients who received CUS, the location of DVT was more often distal and bilateral in COVID-19 patients compared to non COVID [23], and the diagnosis was obtained within 24 hours in 81% of cases. In a recent study on ICU e non ICU patients with COVID-19 the cumulative proportion of VTE-free patients in the medical wards at 7, 14, 21, and >21 days was 91%, 80.2%, 62.4%, and 62.4%, respectively [24]. Taken together these results seem to point out that the occurrence of DVT is not confined to a specific time-frame, and that ultrasound monitoring should be considered during the entire hospital stay.

Patients diagnosed with DVT had significantly higher values of the median peak D-dimers (5403 vs 1723 ng/mL; p = 0.004) with respect to those without DVT. A peak D-dimers value of 2000 ng/mL was the most accurate cut-off to identify patients with DVT with a high negative predictive value and significant sensitivity (AUC 0.703; 95% CI 0.572–0.834; p = 0.004, sensitivity 75%, specificity 65%, negative predictive value 93%, positive predictive value 21%, accuracy 58%). At multivariable analysis patients with D-dimer peak levels greater than 2000 ng/ml had a relative risk of DVT 3.7 times greater than patients with values below that limit (RR 3.74; 95%CI 1.27–10, p = 0.016). Similarly, in a retrospective French cohort study of patients with COVID-19 hospitalized in medical wards, receiving prophylaxis with 40 mg of enoxaparin daily, D-dimer levels measured at hospital admission predicted VTE, and the negative predictive value of D-dimer levels < 1000 ng/ml for DVT diagnosis was 90% [24]. Actually, all these data suggest that serial D-dimer monitoring could improve risk estimate of DVT in acutely ill COVID-19 patients. Nevertheless, special caution should be reserved when considering D-dimers in the diagnosis of VTE due to the well-known limitations in sensitivity, specificity, and overall accuracy of the method, as well as the undefined best strategy for D-dimer evaluation (i.e. single cut-point or dynamic change).

Finally, in this study we did not find any difference in DVT incidence when different anticoagulant doses were compared, although a difference in mean daily dose of enoxaparin was observed. This latter finding even if statistically significant, does not appear clinically relevant. Patterns of anticoagulation in COVID-19 are quite variable and heterogeneous at different facilities, and the optimal regimen of thromboprophylaxis in patients with COVID-19 is still matter of research. A potential for heparin in decreasing mortality through reduction of fatal thrombotic events and the anti-inflammatory effects has been postulated [25, 26]. However, if higher than standard anticoagulant doses have a consistent net clinical benefit in reducing mortality is still largely unknown [27]. Currently, many large international trials, with the aim to explore the optimal anticoagulant doses in COVID-19 patients are ongoing [28] and the very recent recommendations rely more on consensus of experts than on evidence [16].

A point to be discussed is the lack of association of DVT diagnosis and clinically relevant outcome measures, such as in-hospital mortality and ICU transfer, so that we cannot recommend a screening evaluation of DVT in COVID-19 patients based on these results. Nonetheless, at the same time we cannot exclude that, since about all DVTs were clinically silent, in the absence of ultrasound surveillance they would have been undetected and subsequently undertreated, with likely relevant consequences. This is unmet clinical need should be assessed with a cohort of COVID patients not screened for DVT, as controls, in order to evaluate if an ultrasonographic diagnosis (and the subsequent shift to a higher intensity of anticoagulation) is associated with different prognosis. It is worth to note that very recently Raskob et al., in a secondary analysis of the MAGELLAN trial [29], clearly showed the importance of asymptomatic

proximal DVT found by CUS screening in acutely ill medical patients (including those with pneumonia/sepsis) in predicting all cause mortality which was double compared with patients without VTE.

This study has some limitations. First, the 3-point CUS protocol was intended to detect proximal DVT and did not include the extension of US to whole leg. The reason for this choice was to obtain reliable data that could have general impact, since in most IMUs there is a staff trained and expert in performing CUS of proximal venous district, while the examination of the infrapopliteal veins require more expertise and is affected by a higher rate of false positive and false negative results. Nonetheless, an underestimation of DVT incidence of the distal district cannot be ruled out. Second, we cannot exclude that the lack of standardization of anticoagulant doses, caused by the absence of definitive evidence [30] could have had some impact on DVT occurrence. Third, the value and timing to which to refer for D-dimer determination has not been standardized and we referred to the peak value nearest to DVT, and the highest absolute value for patients without DVT. This is an arbitrary definition, but reviewing similar articles on the topic, we appreciated that the optimal strategy for D-dimer testing is still largely undefined [16, 31, 32]. Fourth, CTPA was not mandatory in all patients, and then we cannot establish the contribution of isolated pulmonary embolism to the overall incidence of VTE.

In conclusion, the findings of this multicenter study confirm the high incidence of lower limb DVT despite anticoagulation in patients with COVID-19 pneumonia admitted to Internal Medicine wards. A surveillance protocol by serial CUS of the lower limbs is a simple, rapid and inexpensive tool to timely identify DVT that would go otherwise largely undetected. In the presence of older age and D-dimer levels exceeding 2000 ng/mL a high suspicion of DVT should be raised. Strategies to better define risk stratification of DVT and its timely identification with ultrasonographic surveillance should be further investigated.

## Author Contributions

**Conceptualization:** Filippo Pieralli, Fulvio Pomero, Rossella Marcucci, Antonio Mancini, Andrea Ungar, Alberto Fortini.

**Data curation:** Margherita Giampieri, Fabio Luise, Antonio Mancini.

**Formal analysis:** Filippo Pieralli, Fulvio Pomero.

**Investigation:** Alessandro Milia, Lucia Sammicheli, Irene Tassinari, Francesca Caldi, Francesca Innocenti, Antonio Faraone, Chiara Beltrame.

**Methodology:** Francesca Innocenti, Antonio Faraone, Andrea Ungar.

**Supervision:** Filippo Pieralli, Domenico Prisco.

**Writing – original draft:** Filippo Pieralli, Fulvio Pomero, Margherita Giampieri, Fabio Luise, Alberto Fortini.

**Writing – review & editing:** Filippo Pieralli, Fulvio Pomero, Rossella Marcucci, Domenico Prisco, Riccardo Pini, Alberto Fortini.

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
