## [Decision Letter · Decision Letter 0]

31 Mar 2021

PONE-D-21-08405

Incidence of DVT through an ultrasound surveillance protocol in patients with COVID-19 pneumonia in non-ICU setting: a multicenter prospective study

PLOS ONE

Dear Dr. Pieralli,

Thank you for submitting your manuscript to PLOS ONE. After careful consideration, we feel that it has merit but does not fully meet PLOS ONE’s publication criteria as it currently stands. Therefore, we invite you to submit a revised version of the manuscript that addresses the points raised during the review process.

We look forward to receiving your revised manuscript.

Kind regards,

Aleksandar R. Zivkovic

Academic Editor

PLOS ONE

2. Please include the full name of the ethics committee that approved your study in the manuscript Methods.

3. In your ethics statement in the manuscript and in the online submission form, please ensure that you have discussed whether all data/samples were fully anonymized before you accessed them and/or whether the IRB or ethics committee waived the requirement for informed consent. If patients provided informed written consent to have data/samples from their medical records used in research, please include this information.

Reviewers' comments:

Reviewer #1: Pieralli et al conducted a prospective study to determine the incidence of LE DVT in hospitalized non-ICU COVID-19 patients using a Lower extremity Doppler CUS surveillance protocol. Although there have been multiple previous reports on this topic in the literature, strengths of this report is that it represents a fairly large multicenter and prospective study on the topic with interesting conclusions, namely that the incidence of unsuspected LE DVT, about 50% proximal, is still quite high (13.7%) despite standard or escalated-dose thromboprophylaxis. There are however, a number of issues that need to be addressed:

1. For the Introduction, the second paragraph seems contradictory, the authors cite evidence in non-ICU patients, followed ny a sentence that describes "studies carried out in ICU patients". This needs clarification.

2. Under Methods, more detail is needed for the Doppler CUS exam. Was it standardized across the 3 hospitals? Was it formalized into a standardized 5-point exam when applicable or a 2 point bedside exam if the patient was ill? As D-dimer is an important biomarker as stated by these and other authors, I would state the units used and the upper limit of normal across all 3 hospitals.

3. For Statistical Analysis, I would be consistent and use either mean or median for lab values, but not either/or.

4 Under Results, at least in US healthcare settings, COVID-19 patients requiring non-invasive mechanical ventilation are considered requiring ICU level-of-care, even if they are not in a named ICU. This may represent a major limitation of the manuscript in my view, as at least 16% of patients would be considered ICU patients (not medical ward).

5. For the D-dimer results, literature now supports their relative value as prognostic tools in COVID-19 rather than absolute value, see recent large cohort of >9400 COVID-19 patients from a NY area health system (Cohen S et al Thromb Haemost 2021) that confirms the initial observations from Tang et al JTH 2020) of Dd > 4X or 6 X ULN. I would describe all Dd results relative to the ULN, using local laboratory normal.

6. Under Discussion, authors neglect one of the more important and relevant recent publications in the field (Raskob et al JAHA 2021) that clearly describes the importance of asymptomatic proximal DVT found by screening Doppler LE CUS in medical patients (including those with pneumonia/sepsis) in predicting all cause mortality. This points to the relevance of using screening ultrasonography. In addition on page 15, the authors should provide more background as to the randomized controlled trials of thromboprophylaxis in hospitalized COVID-19 patients that are using screening LE ultrasonography as part of the study's composite endpoint, thus adding weight to their study's findings.

7. Also under Discussion, page 10, the authors spend too much time in Discussing the relevance of their finding based on the PPS. The study is woefully underpowered to make any association between the authors findings and thromboprophylaxis regimens as well as characteristics of the PPS. The sample size needed would be ~10,000 COVID-19 patients.

8. Lastly, I would have the authors spend more time in the Discussion comparing their findings relative to other studies, and the relative strength of their data compared to previous work. They have only set aside a single paragraph for this, page 12 Paragraph 2. The authors spend more time discussing their aspects of their work sucg as Dd, timing, thromboprophylaxis regimens etc which the study is underpowered to detect meaningful differences.

Minor comments:

1. English grammar needs slight improvement throughout the manuscript.

2. The statement on the MEDENOX trial in the Discussion on page 13 is comparing apples to oranges. MEDENOX used venographic detection, which characteristically captures more LE DVT than ultrasonograpic detection. A more appropriate comparator would by the PREVENT trial published in Circulation 2004.

Reviewer #2: This is an interesting paper focused on DVT incidence in non-ICU COVID 19 patients.

To date, it is the largest series about this topic in non critically-ill patients studied with serial ultrasound scan.

It is a well written multicenter study with valuable data. However, information is similar to other already published papers an no new and very original data is included.

In general, my main concern regards to the absence of a complete ultrasound scan, including distal veins, to all of the patients (42.5% of patients’ distal veins were not scanned). This may represent a detection bias. As explained by the authors, sonographers were not experts in this distal study.

In addition, a sample size was not calculated and result obtained is not precise.

However, data about proximal DVT is valuable as well.

Specific comments:

KEYWORDS

. Keywords are well selected but I would suggest adding the MESH term “Ultrasonography”.

ABSTRACT

. Ultrasound scan focused on detecting DVT could be done using two approaches: “2 points compression ultrasound” or “whole leg ultrasound”. Please, describe the technique used.

. Methods must describe “design” (I suppose it is a “cohort” study), “inclusion and exclusion criteria” (what was the criteria used to admit patients to the hospital?; did you consider any exclusion? (cancer patients or previous history of VTE, for instance). In addition, you must describe “outcomes measured”, not only doses of anticoagulation used (for instance, DDimer, described in results). In my view, it is more interesting to know the protocol used to prescribe different doses of heparin than the description of doses used.

. In addition, doses of heparin may vary during the whole length of stay. Did you consider these changes of doses?

. Methods must include the planned length of the surveillance.

. Did you calculate the sample size?

. What is the definition of “peak DDimer level”?

. Regarding the conclusion, I am not totally sure that a serial CUS surveillance is useful for these patients. This is a protocol to determine incidence but benefits of the surveillance is not proven with this design.

INTRODUCTION

. Patients with pneumonia could be admitted to other medical departments, such as Pneumology. In my view, this paper regards to patients admitted to non-ICU departments, such as Pneumology, not only Internal Medicine Units.

. Regarding the objective, the incidence of DVT in non-ICU patients, in my view, is not an unresolved issue. To date, there are some papers focused on this issue and some systematic reviews. For instance:

Proximal deep vein thrombosis and pulmonary embolism in COVID-19 patients: a systematic review and meta-analysis. Longchamp G, Manzocchi-Besson S, Longchamp A, Righini M, Robert-Ebadi H, Blondon M.Thromb J. 2021 Mar 9;19(1):15. doi: 10.1186/s12959-021-00266-x

In this paper, 8 papers are described on non-ICU patients with screening ultrasound.

METHODS

. IS the design a cohort study?

. Did you consider any exclusion criteria? For instance, patients with known previous DVT.

. Reference 11 regards to recommendations about prophylaxis in COVID19 patients. A brief summary would be interesting to allow readers a better access to this information.

. Who performed the CUS? Were all investigators experts in ultrasonography?

. “The ultrasound scan was performed along the proximal femoral and popliteal district bilaterally, and if possible was extended to the distal infrapopliteal vein district”. In which circumstances it is not possible? Please describe

. How “incident diagnosis” was defined? Did you exclude “prevalent diagnosis”? I mean patients with in-hospital already known DVT or previous DVT.

. “Incidence” could be defined as “cumulative incidence” or “incidence rate”. Please define the exact term measured.

RESULTS

. Overall incidence was 13.7%. However, we need to know a dispersion value, such as the 95%CI. (in this case, 9.2% to 18.2%, in my view, too wide).

. We would like to know the reason why 42.5% explorations did not include infrapopliteal veins.

. How did you select the most accurate cut-off of peak DDimer? Did you calculate the Youden’s J-statistic?

DISCUSSION

. This is a well written discussion and I have no comments about it.

TABLES

. Table 1: More information about DDimer values would be interesting. For instance, percentage of different cut-off values (% of DDimer>500, DDimer>1000, DD>2000, ...).

Reviewer #3: Thank you very much for the opportunity to review this manuscript. It is important to continue to produce evidence concerning SARS-CoV2.

I do believe there is some points that have to be improved.

1. Introduction:

- The authors should provide more evidence there is a different incidence of DVT in wards and ICU. Does this occurs in other disease?

2. Methods

-Did you performed CUS in patients that were transferred to the ICU? Please clarify this in the methods.

- Padua prediction score should be better explained (maximum points, cut point for severe, moderate, mild disease)

- I understand that incidence is about new cases. Ideally you should have a negative test to start with and then a follow up. I think your study is a period prevalence.

3. Results:

- How many patients that had DVT had a CTPA?

- I sugest data from PE and CTPA be included in table 2

- Do you have the duration of COVID19 symptoms before hospital admission?

4. Discussion:

- You should discuss age as risk factor for DVT. If this should have an impact on how we manage the elderly.

- The finding about sequential ultrasound is a nice one. Please elaborate more.

- Overall I believe the discussion can be enriched with a more robust review of literature. I lot has been published recently about this topic.

I add 3 suggestion of studies, but there are others:

DOI: 10.1016/j.thromres.2020.10.012

DOI: 10.1007/s11239-021-02395-6

https://doi.org/10.1371/journal.pone.0245565

6. PLOS authors have the option to publish the peer review history of their article (what does this mean?). If published, this will include your full peer review and any attached files.

Reviewer #1: No

Reviewer #2: **Yes: **Sergi Bellmunt-Montoya

Reviewer #3: No

---

## [Author Response · Author response to Decision Letter 0]

4 May 2021

Journal requirements.

Please include the full name of the ethics committee that approved your study in the manuscript Methods.

In your ethics statement in the manuscript and in the online submission form, please ensure that you have discussed whether all data/samples were fully anonymized before you accessed them and/or whether the IRB or ethics committee waived the requirement for informed consent. If patients provided informed written consent to have data/samples from their medical records used in research, please include this information.

The following paragraph has been added to methods.

Human beings were involved in the research, the study protocol was approved by the ethics committee of the coordinating center, Azienda Ospedaliera Universitaria Careggi, Florence, Italy (COCORA protocol n17104), and was performed in agreement with the principles set in the Declaration of Helsinki. Informed consent was not needed since point-of-care ultrasonography is a standard of care for the evaluation and monitoring of patients with COVID-19; only signed consent for personal data collection and treatment was requested; all the data were analyzed anonymously.

Responses to Reviewer #1

We are grateful for the positive comments and useful suggestions. The answers to the queries are reported below. Moreover modifications to the text, that eventually have been incorporated in the revised version of the manuscript, are in bold characters. 

1. For the Introduction, the second paragraph seems contradictory, the authors cite evidence in non-ICU patients, followed ny a sentence that describes "studies carried out in ICU patients". This needs clarification.

……… The paragraph in introduction has been modified in: “Actually, the incidence of VTE in non-ICU patients admitted for infectious respiratory disease is estimated around 5%, with adequate thromboprophylaxis, while the incidence in COVID-19 patients has been reported as high as 20% [4-11]. In the SARS-CoV-2 pandemic scenario, the most reliable data derive from studies carried out in ICU patients, while less robust information is available for patients with COVID-19 pneumonia admitted to non-ICU settings, such as those of Pneumology and Internal Medicine Units (IMUs)”, and has been incorporated in the manuscript

2. Under Methods, more detail is needed for the Doppler CUS exam. Was it standardized across the 3 hospitals? Was it formalized into a standardized 5-point exam when applicable or a 2 point bedside exam if the patient was ill? As D-dimer is an important biomarker as stated by these and other authors, I would state the units used and the upper limit of normal across all 3 hospitals.

The method used for DVT detection has been clarified and the following sentence has been added in methods section.

“The ultrasound scan was performed along the proximal femoral and popliteal district bilaterally, by a three-point examination of the common and superficial femoral veins, and the popliteal veins [U.M. Hamper, M.R. DeJong, L.M. Scoutt. Ultrasound evaluation of the lower extremity veins. Radiol Clin North Am, 45 (2007), pp. 525-547], and if possible was extended to the distal infrapopliteal vein district.”. 

The method was in use in all the three hospitals according to a widely adopted 3-point protocol in use in clinical practice [U.M. Hamper, M.R. DeJong, L.M. Scoutt. Ultrasound evaluation of the lower extremity veins. Radiol Clin North Am, 45 (2007), pp. 525-547]. The evaluation of the infrapopliteal veins was not standardized and left to the discretion and the expertise of the examiner. 

I would state the units used and the upper limit of normal across all 3 hospitals. 

The units for D-dimer were expressed as ng/mL FEU (fibrinogen equivalent unit) with an upper limit of 500 ng/mL FEU. The method and the laboratory normal values were the same for all the three hospitals’ laboratories. Methods are now implemented as you suggested, see text.

3. For Statistical Analysis, I would be consistent and use either mean or median for lab values, but not either/or. Modified as suggested.

4. Under Results, at least in US healthcare settings, COVID-19 patients requiring non-invasive mechanical ventilation are considered requiring ICU level-of-care, even if they are not in a named ICU. This may represent a major limitation of the manuscript in my view, as at least 16% of patients would be considered ICU patients (not medical ward).

We thank the reviewer by pointing out a crucial issue in the management of severe COVID-19 pneumonia in the real context in a significant part of European countries. Ideally, all patients with severe COVID-19 requiring non invasive ventilation or high flow O2 oxygenation should be assisted in the ICU setting. Unfortunately, in the real world context of many European countries where public universal health-care systems are in place, the numbers of ICU beds with respect to total hospital capacities is far less than that available in North America. In Europe, the estimates of ICU beds is on average 11.5 beds per 100.000 inhabitants, with a wide range varying from 29.2/100.000 in Germany to 4.2/100.000 in Portugal [Rhodes A, Ferdinande P, Moreno RP. The variability of critical care bed numbers in Europe- Intensive Care Medicine, 2012; 38: 1647-1653]. In Italy and France the estimate is of 12.5/100.000 inhabitants, featuring nearly 4% ICU beds as percentage of acute care beds. These figures, on average, are more than 1/3 less the capacity in the US, and it is a matter of fact that it represents a well known limit of the universal coverage health care systems. During the COVID-19 pandemic that critical point has been particularly stressed and non invasive ventilation or HFNC has been frequently delivered outside the ICU setting, mainly in Internal Medicine units. From a recent survey during COVID-19 pandemic in Italy, a variable proportion of patients ranging from 30 to 50%, received helmet CPAP or other forms of non invasive ventilation when cared for COVID pneumonia in Internal Medicine wards [Montagnani, A., Pieralli, F., Gnerre, P. et al. COVID-19 pandemic and Internal Medicine Units in Italy: a precious effort on the front line. Intern Emerg Med 15, 1595–1597 (2020). https://doi.org/10.1007/s11739-020-02454-5. Coppadoro A, Benini A, Fruscio R, et al. Helmet CPAP to treat hypoxic pneumonia outside the ICU: an observational study during the COVID-19 outbreak. Crit Care. 2021;25(1):80. Published 2021 Feb 24. doi:10.1186/s13054-021-03502-y], and similar figures are reported in other European countries [Alviset S, Riller Q, Aboab J, et al. Continuous Positive Airway Pressure (CPAP) face-mask ventilation is an easy and cheap option to manage a massive influx of patients presenting acute respiratory failure during the SARS-CoV-2 outbreak: A retrospective cohort study. PLoS One. 2020;15(10):e0240645. Published 2020 Oct 14. doi:10.1371/journal.pone.0240645]. For these reasons, we believe that the analysis of these real world data is very useful and represents a point strength of our paper, and not a limitation. 

5. For the D-dimer results, literature now supports their relative value as prognostic tools in COVID-19 rather than absolute value, see recent large cohort of >9400 COVID-19 patients from a NY area health system (Cohen S et al Thromb Haemost 2021) that confirms the initial observations from Tang et al JTH 2020) of Dd > 4X or 6 X ULN. I would describe all Dd results relative to the ULN, using local laboratory normal. We thank the reviewer for this observation; however, since the method and the upper limit for D-dimers were the same across all the 3 hospitals, we considered more informative the results expressed as ng/ml FEU. D-dimers expressed as quartiles are now reported in Table.

6. Under Discussion, authors neglect one of the more important and relevant recent publications in the field (Raskob et al JAHA 2021) that clearly describes the importance of asymptomatic proximal DVT found by screening Doppler LE CUS in medical patients (including those with pneumonia/sepsis) in predicting all cause mortality. This points to the relevance of using screening ultrasonography. In addition on page 15, the authors should provide more background as to the randomized controlled trials of thromboprophylaxis in hospitalized COVID-19 patients that are using screening LE ultrasonography as part of the study's composite endpoint, thus adding weight to their study's findings.

We thank the reviewer for the useful suggestions. The study by Raskob et al. was not published at the time of the first submission to PlosOne. We read with interest that paper which adds information on the prognostic implications of detecting asymptomatic DVT in critically ill patients. As suggested by the reviewer this adds relevance to the use of screening ultrasonography in the critically ill patient, supporting the use of a ultrasound surveillance protocol in patients with COVID-19 pneumonia. We added the following paragraph and relative reference in discussion. “It is worth to note that very recently Raskob et al., in a secondary analysis of the MAGELLAN trial (NCT00571649), clearly showed the importance of asymptomatic proximal DVT found by CUS screening in acutely ill medical patients (including those with pneumonia/sepsis) in predicting all cause mortality which was double compared with patients without VTE [Raskob GE, Spyropoulos AC, Cohen AT, et al. Association Between Asymptomatic Proximal Deep Vein Thrombosis and Mortality in Acutely Ill Medical Patients. J Am Heart Assoc. 2021;10(5):e019459. doi:10.1161/JAHA.120.019459].

7. Also under Discussion, page 10, the authors spend too much time in Discussing the relevance of their finding based on the PPS. The study is woefully underpowered to make any association between the authors findings and thromboprophylaxis regimens as well as characteristics of the PPS. The sample size needed would be ~10,000 COVID-19 patients.

Based on the reviewer’s suggestions, the discussion has been reduced accordingly.

8. Lastly, I would have the authors spend more time in the Discussion comparing their findings relative to other studies, and the relative strength of their data compared to previous work. They have only set aside a single paragraph for this, page 12 Paragraph 2. The authors spend more time discussing their aspects of their work such as Dd, timing, thromboprophylaxis regimens etc which the study is underpowered to detect meaningful differences.

According to your suggestions the discussion has been modified.

Minor comments:

1. English grammar needs slight improvement throughout the manuscript.

The manuscript was revised in order to improve English language grammar and flow.

2. The statement on the MEDENOX trial in the Discussion on page 13 is comparing apples to oranges. MEDENOX used venographic detection, which characteristically captures more LE DVT than ultrasonograpic detection. A more appropriate comparator would by the PREVENT trial published in Circulation 2004.

According to your useful suggestions we modified the discussion as follows.

“In PREVENT trial, which compared dalteparin 5000 IU daily to placebo for the prevention of venous thromboembolism in acutely ill medical patients, the incidence of proximal DVT mainly assessed by ultrasonography was 2% in the active treatment arm and 4.05% in the placebo group [18]. Indeed, the incidence of DVT in COVID-19 patients on anticoagulant therapy is much higher than the incidence of hospitalized medically ill non COVID-19 patients not receiving prophylactic treatment.”

Responses to Reviewer #2

We are grateful for the positive comments and useful suggestions. The answers to the queries are reported below. Moreover modifications to the text, that eventually have been incorporated in the revised version of the manuscript, are in bold characters. 

Specific comments:

KEYWORDS. Keywords are well selected but I would suggest adding the MESH term “Ultrasonography”.

As suggested. Ultrasonography has been added to keywords.

Ultrasound scan focused on detecting DVT could be done using two approaches: “2 points compression ultrasound” or “whole leg ultrasound”. Please, describe the technique used.

The method used for DVT detection has been clarified and the following sentence has been added in methods section.

“The ultrasound scan was performed along the proximal femoral and popliteal district bilaterally, by a three-point examination of the common and superficial femoral veins, and the popliteal veins [U.M. Hamper, M.R. DeJong, L.M. Scoutt. Ultrasound evaluation of the lower extremity veins. Radiol Clin North Am, 45 (2007), pp. 525-547], and if possible was extended to the distal infrapopliteal vein district.” CUS examination was performed by experienced physicians with long-standing expertise in first-level vascular ultrasonography; however not all physicians were able to reliably perform an infrapopliteal venous scanning.

Methods must describe “design” (I suppose it is a “cohort” study), “inclusion and exclusion criteria” (what was the criteria used to admit patients to the hospital?; did you consider any exclusion? (cancer patients or previous history of VTE, for instance). 

The term cohort has been added to methods: “This is a prospective cohort study…..”. Since the observational nature of the study neither inclusion nor exclusion criteria were considered. No patient was excluded from the enrollment. 

In addition, you must describe “outcomes measured”, not only doses of anticoagulation used (for instance, DDimer, described in results). In my view, it is more interesting to know the protocol used to prescribe different doses of heparin than the description of doses used.

. In addition, doses of heparin may vary during the whole length of stay. Did you consider these changes of doses?

We agree with the Reviewer that heparin doses may vary during the whole length of stay; in fact we initially considered variations in dosage, but we observed infrequently modifications of the doses initially prescribed, unless the occurrence of VTE. For this reason we decided to classify doses as low, intermediate, high (as reported in methods) according to literature criteria [Marietta M, Ageno W, Artoni A, et al. COVID-19 and haemostasis: a position paper from Italian Society on Thrombosis and Haemostasis (SISET). Blood Transfus. 2020;18(3):167-169. doi:10.2450/2020.0083-20]. That way to express heparin doses seems more adherent to clinical practice (LMWH is in fixed doses) than express mean daily doses, that however were calculated and reported in table 2.

Methods must include the planned length of the surveillance.

As reported in the method section all patients were monitored for DVT during the hospital stay until discharge. “All patients were screened for DVT of the lower limbs with serial color-coded Doppler and compression ultrasonography (CUS) within 72 hours since admission and subsequently at 5-7 day intervals, and before discharge.. 

Did you calculate the sample size?

The sample size was not calculated since this was an observational and not an interventional study.

.What is the definition of “peak DDimer level”?

As already reported in the methods section, for the purposes of this study “peak D-dimers” was arbitrarily defined as the highest value during hospital stay for patients without DVT and the highest value at the time (±24h) DVT was diagnosed for the DVT group.

Regarding the conclusion, I am not totally sure that a serial CUS surveillance is useful for these patients. This is a protocol to determine incidence but benefits of the surveillance is not proven with this design.

We agree with the reviewer that our study cannot offer ultimate information on benefits on clinical outcome with a DVT surveillance protocol. In fact, the study was not designed to answer that question and for that purpose it is certainly underpowered. To test the benefit on clinical “hard” end-points of a surveillance protocol of DVT probably the numbers would be in the range of 8.000-10.000 patients. However this is a relevant issue and, to give more information as suggested also by reviewer N.1, we modified discussion adding data from Magellan trial. 

“A point to be discussed is the lack of association of DVT diagnosis and clinically relevant outcome measures, such as in-hospital mortality and ICU transfer, so that we cannot recommend a screening evaluation of DVT in COVID-19 patients based on these results. Nonetheless, at the same time we cannot exclude that, since about all DVTs were clinically silent, in the absence of ultrasound surveillance they would have been undetected and subsequently undertreated, with likely relevant consequences. This is an unmet clinical need that should be assessed with a cohort of COVID patients not screened for DVT, as controls, in order to evaluate if an ultrasonographic diagnosis (and the subsequent shift to a higher intensity of anticoagulation) is associated with better prognosis. It is worth to note that very recently Raskob et al., in a secondary analysis of the MAGELLAN trial (NCT00571649), clearly showed the importance of asymptomatic proximal DVT found by CUS screening in acutely ill medical patients (including those with pneumonia/sepsis) in predicting all cause mortality which was double compared with patients without VTE [Raskob GE, Spyropoulos AC, Cohen AT, et al. Association Between Asymptomatic Proximal Deep Vein Thrombosis and Mortality in Acutely Ill Medical Patients. J Am Heart Assoc. 2021;10(5):e019459. doi:10.1161/JAHA.120.019459].” 

INTRODUCTION

 Patients with pneumonia could be admitted to other medical departments, such as Pneumology. In my view, this paper regards to patients admitted to non-ICU departments, such as Pneumology, not only Internal Medicine Units.

Regarding the objective, the incidence of DVT in non-ICU patients, in my view, is not an unresolved issue. To date, there are some papers focused on this issue and some systematic reviews. For instance:

Proximal deep vein thrombosis and pulmonary embolism in COVID-19 patients: a systematic review and meta-analysis. Longchamp G, Manzocchi-Besson S, Longchamp A, Righini M, Robert-Ebadi H, Blondon M.Thromb J. 2021 Mar 9;19(1):15. doi: 10.1186/s12959-021-00266-x. In this paper, 8 papers are described on non-ICU patients with screening ultrasound.

According to your suggestions, introduction has been modified as follows, including the suggested reference (Longchamp et al.) which was not published at the time of submission): 

In the SARS-CoV-2 pandemic scenario, the most reliable data derive from studies carried out in ICU patients, while less robust information is available for patients with COVID-19 pneumonia admitted to non-ICU settings, such as those of Pneumology and Internal Medicine Units (IMUs).

METHODS

. IS the design a cohort study?

. Did you consider any exclusion criteria? For instance, patients with known previous DVT.

. Reference 11 regards to recommendations about prophylaxis in COVID19 patients. A brief summary would be interesting to allow readers a better access to this information.

. Who performed the CUS? Were all investigators experts in ultrasonography?

. “The ultrasound scan was performed along the proximal femoral and popliteal district bilaterally, and if possible was extended to the distal infrapopliteal vein district”. In which circumstances it is not possible? Please describe

. How “incident diagnosis” was defined? Did you exclude “prevalent diagnosis”? I mean patients with in-hospital already known DVT or previous DVT.

“Incidence” could be defined as “cumulative incidence” or “incidence rate”. Please define the exact term measured.

The incidence is defined as cumulative incidence; the term has been more exactly defined in methods.

RESULTS

. Overall incidence was 13.7%. However, we need to know a dispersion value, such as the 95%CI. (in this case, 9.2% to 18.2%, in my view, too wide).

. We would like to know the reason why 42.5% explorations did not include infrapopliteal veins.

The explanations to your query is reported in methods where, by protocol, the US was obtained with a 3-points venous scanning, the decision to explore the infrapopliteal district (which require more expertise) was left to the physician performing the examination. This is reported in methods.

. How did you select the most accurate cut-off of peak DDimer? Did you calculate the Youden’s J-statistic?

The most accurate cut-off for D-dimer has been calculated by ROC curve analysis and the Youden index as you can find in methods and results. Youden index can easily be calculated as 0.30.

DISCUSSION

. This is a well written discussion and I have no comments about it.

We thank the reviewer for the positive comment.

TABLES

. Table 1: More information about DDimer values would be interesting. For instance, percentage of different cut-off values (% of DDimer>500, DDimer>1000, DD>2000, ...).

Table 1 was modified reporting D-dimer quartiles accordingly.

Reviewer #3: Thank you very much for the opportunity to review this manuscript. It is important to continue to produce evidence concerning SARS-CoV2.

I do believe there is some points that have to be improved.

1. Introduction:

- The authors should provide more evidence there is a different incidence of DVT in wards and ICU. Does this occurs in other disease?

The introduction has been modified upon suggestions of the reviewers. The paragraph in introduction has been modified in: “Actually, the incidence of VTE in non-ICU patients admitted for infectious respiratory disease is estimated around 5%, with adequate thromboprophylaxis, while the incidence in COVID-19 patients has been reported as high as 20% [4-11]. In the SARS-CoV-2 pandemic scenario, the most reliable data derive from studies carried out in ICU patients, while less robust information is available for patients with COVID-19 pneumonia admitted to non-ICU settings, such as those of Internal Medicine Units (IMUs)”, and has been incorporated in the manuscript.

2. Methods

-Did you performed CUS in patients that were transferred to the ICU? Please clarify this in the methods.

CUS was performed at admission and at 5-7 day intervals until discharge or ICU transfer.

- Padua prediction score should be better explained (maximum points, cut point for severe, moderate, mild disease)

- I understand that incidence is about new cases. Ideally you should have a negative test to start with and then a follow up. I think your study is a period prevalence.

The incidence is defined as cumulative incidence; the term has been more exactly defined in methods.

3. Results:

- How many patients that had DVT had a CTPA?

- I sugest data from PE and CTPA be included in table 2

The results pertaining to PE are reported in the results section. “Overall, 9 patients (3.9%) were diagnosed with PE at CTPA. Of these, 5 had a concomitant DVT of the lower limbs, while in 4 cases PE was isolated. CTPA was available in 67 out of 227 patients (29.5%). 

Do you have the duration of COVID19 symptoms before hospital admission?

This information is not available for this cohort.

4. Discussion:

- You should discuss age as risk factor for DVT. If this should have an impact on how we manage the elderly.

- The finding about sequential ultrasound is a nice one. Please elaborate more.

- Overall I believe the discussion can be enriched with a more robust review of literature. I lot has been published recently about this topic.

We thank the reviewer for comments. Other reviewers suggested similar comments, then the discussion has been modified accordingly, including new references too. On this last point we should note, however, that literature on COVID is overwhelmed by rapidly evolving literature and at the time of the first submission of the manuscript some relevant articles on the topic were not published. 

I add 3 suggestion of studies, but there are others:

We thank the reviewer for suggestions; the more relevant studies have been incorporated in the manuscript and references.

---

## [Editor Report · Decision Letter 1]

7 May 2021

Incidence of DVT through an ultrasound surveillance protocol in patients with COVID-19 pneumonia in non-ICU setting: a multicenter prospective study

PONE-D-21-08405R1

Dear Dr. Pieralli,

We’re pleased to inform you that your manuscript has been judged scientifically suitable for publication and will be formally accepted for publication once it meets all outstanding technical requirements.

Kind regards,

Aleksandar R. Zivkovic

Academic Editor

PLOS ONE

---

## [Editor Report · Acceptance letter]

12 May 2021

PONE-D-21-08405R1 

Incidence of deep vein thrombosis through an ultrasound surveillance protocol in patients with COVID-19 pneumonia in non-ICU setting: a multicenter prospective study. 

Dear Dr. Pieralli:

I'm pleased to inform you that your manuscript has been deemed suitable for publication in PLOS ONE. Congratulations! Your manuscript is now with our production department. 

Kind regards, 

on behalf of

Dr. Aleksandar R. Zivkovic 

Academic Editor

PLOS ONE